# Genetic Characterization of a Sheep Population in Oaxaca, Mexico: The Chocholteca Creole

**DOI:** 10.3390/ani11041172

**Published:** 2021-04-20

**Authors:** Teodulo Salinas-Rios, Jorge Hernández-Bautista, Araceli Mariscal-Méndez, Magaly Aquino-Cleto, Amparo Martínez-Martínez, Héctor Maximino Rodríguez-Magadán

**Affiliations:** 1Facultad de Medicina Veterinaria y Zootecnia, Universidad Autónoma Benito Juárez de Oaxaca, Oaxaca 68110, Mexico; salinas980@hotmail.com (T.S.-R.); jorgeherba@hotmail.com (J.H.-B.); mariscalma@hotmail.com (A.M.-M.); mvzmagalyaquino@hotmail.com (M.A.-C.); 2Departamento de Genética, Universidad de Córdoba, 14014 Córdoba, Spain; animalbreedingconsulting@gmail.com

**Keywords:** consanguinity, Chocholteca Creole, microsatellites, *Ovis aries*, populational substructure

## Abstract

**Simple Summary:**

Creole animals are an important genetic resource due to their adaptation to adverse conditions, however, many of them have disappeared or have been reduced in number; therefore, their identification and preservation should be promoted. In the Mixteca zone of Oaxaca, México, there is a group of creole sheep with physical differences with respect to the other known breeds. The objective of the present study was to determine the degree of differentiation among the individuals of the population of creole sheep and the similarities with other breeds, as well as to measure their level of conservation and consanguinity. It was found that there is a group of sheep which is different from the presently known breeds and that they do not present high degree of consanguinity, and thus may be considered a new breed. Therefore, it is proposed that they are identified as the Chocholteca Creole breed, in honor of the ethnic group which inhabits this region. The present study is very important because it discovers a new genotype of sheep, which amplifies the genetic diversity. Therefore, further studies are needed, given that they are a potential alternative of meat and wool.

**Abstract:**

Creole sheep in México have undergone crossbreeding, provoking the loss of genetic variability. The objective of the present study is to determine the intra-racial genetic diversity, the genetic relationship with other genotypes, and the populational substructure of the Oaxacan Creole sheep. Twenty-nine blood samples were obtained of Creole sheep of the Oaxaca Mixteca region in México. A genetic analysis was made with 41 microsatellites recommended for studies of genetic diversity in sheep. An analysis was made of genetic diversity, populational structure, and genetic distance with 27 other sheep populations. The study found 205 alleles with a range of 2 to 9 by *locus* and an effective number of 3.33. The intra-racial analysis showed a moderate genetic diversity with values of expected heterozygosity of 0.686 and observed of 0.756, a mean polymorphic information content of 0.609, and a mean coefficient of consanguinity of −0.002. In interracial genetic diversity for the coefficients of consanguinity, the values were F_IS_ = 0.0774, F_IT_ = 0.16993, and F_ST_ = 0.10028, showing an elevated genetic distance with other creole breeds, but close to Argentine Creole, to another Creole of México and the Spanish Merino. Its genetic structure showed that it does not have any populational subdivision nor mixes with the others analyzed. It is concluded that it is a distinct and isolated population and is proposed as the creole breed “Chocholteca” for its conservation.

## 1. Introduction

Sheep farming is one of the principal agricultural activities in the state of Oaxaca, Mexico. The Mixteca region has the highest number of production units [1,2], principally small scale as a system of family subsistence based on pasture grazing [3]. The flocks present in this region have been crossed with genetically improved commercial breeds to obtain better productive characteristics [4], provoking a crossbreeding and reduction of pure native creole breeds, placing them in a status of threatened species. In recent years, this process has caused the disappearance of 7% of the local breeds [5], as well as the loss of their genetic qualities, that give them resistance to the local environment, to diseases and allows them to feed with fodder lacking nutritional quality [6,7].

The High Mixteca zone, of the state of Oaxaca, inhabited by the Chocholteca ethnic group, still conserves a sheep population with no more than 100 animals that present morphological characteristics similar to those introduced over 500 years ago, as well as from other regions of México [8], which over the years became adapted to unfavorable environments and became the base of animal production. The study of this type of creole breeds has become a priority within the programs of sustainable management, in order to prevent their genetic deterioration or possible extinction.

At present, one of the tools most widely used is the characterization of production animals through the employment of molecular markers, such as microsatellites, in order to determine their purity or degree of crossbreeding with other breeds, the genetic diversity within the same population, the level of endogamy, etc. [9,10,11,12,13,14]. Therefore, the objective of the present study is to genetically characterize through microsatellites the Oaxacan Creole sheep, define whether it is a homogeneous breed and distanced from other native Spanish breeds, Creole breeds of América, as well as breeds common in México and Africa. This will make it possible to delineate strategies for recovering its population, as well as to reproduce it and conserve it as a genetic and economic resource for the Oaxacan producers.

## 2. Materials and Methods

### 2.1. Animals, Sampling, and Obtainment of DNA

Twenty-nine creole wool sheep (12 males, 17 females) of the High Mixteca, Oaxaca, México were included. The region is located in the coordinates 17°39′19” longitude west, 91°16′51” latitude north, at an altitude between 2100 and 2500 masl. The climate is humid temperate with rains in summer and an annual temperature of 14 to 20 °C (Figure 1). A sample was taken from each animal of 2 mL of blood from the jugular vein in tubes with EDTA anticoagulant and maintained in ice and transported to the Genetics and Animal Reproduction Laboratory of the FMVZ-UABJO. The samples were taken by a zootechnical veterinarian during the sanitary control of the flock; thus, the approval of an ethics committee was not required.

One-hundred microliters were taken from each sample and placed in filter paper with the individual dates. These were left to dry at room temperature and thus conserved until their analysis in the Applied Molecular Genetics Laboratory of the University of Córdoba, Spain. The DNA of each sample was obtained employing the Walsh technique [15], using the chelate Chelex 100 (Bio-Rad, Hercules, CA, USA) and maintained at −20 °C until its analysis.

### 2.2. Genotyping through Microsatellites

In the analysis, 41 microsatellites were used (BM1818, BM1824, BM6506, BM6526, BM8125, CD5, CMSS66, CRSD247, D5S2, ETH010, ETH225, HSC, ILSTS005, ILSTS008, ILSTS011, ILSTS87, INRA005, INRA006, INRA023, INRA035, INRA049, INRA063, INRA132, INRA172, MAF065, MAF214, McM042, McM527, OarAE129, OarCP20, OarCP34, OarCP49, OarFCB11, OarFCB20, OarFCB304, RM006, SPS113, SPS115, TGLA053, TGLA122, TGLA126), which have been utilized for the identification of other breeds of sheep and are recommended by the Food and Agriculture Organization and the International Society of Animal Genetics (FAO/ISAG). Each microsatellite was amplified by PCR and the obtained fragments were identified by means of 4-capillary electrophoresis in an automatic sequencer ABI 3130XL (Applied Biosystem, Foster City, CA, USA), using the marker “Genescan^®^ 400 HD ROX size Standard” (Thermofisher, Waltham, MA, USA) to corroborate the sizes of the amplified microsatellites. The analysis of fragments and allelic typification were made using the programs Genescan Analysis^®^ 3.1.2 (Fischer Scientific, Waltham, MA, USA) and Genotyper^®^ 2.5.2 (Fischer Scientific), respectively.

### 2.3. Intra-Racial Genetic Diversity

The average number of alleles was determined per locus (MNA), the allelic frequency (Ae), the expected and observed heterozygosis (He), (Ho), along with polymorphic information contained (PIC) using the programs MICROSATELLITE TOOLKIT software of excel [16] and CERVUS 3.0.3 [17]. The effective number of alleles was calculated using the program PopGene [18]; the coefficient of consanguinity, with a confidence interval of 95%, in the program GENETIX v 4.05 [19]; and a Hardy–Weinberg equilibrium test was performed using the program GENEPOP v 3.1c [20], which applies an exact Fisher test by Markov Chain Monte Carlo (MCMC) method [21] and the correction of Bonferroni.

### 2.4. Interracial Genetic Diversity

For the analysis of genetic differentiation of the Oaxacan Creole sheep, it was compared with 27 breeds (Table 1), including some native Spanish, American Creole, African, and breeds common in México. For this purpose, we used the database of the Laboratory of Applied Molecular Genetics of Animal Breeding Consulting S.L. and of the Biovis Consortium (https://biovis.jimdo.com/breeds/ accessed on 21 October 2019). The intellectual property of the data of these breeds belongs to investigators of the network CONBIAND (https://biovis.jimdo.com/investigadores/ accessed on 21 October 2019). The F statistics of Wright [22] were calculated for the coefficient of consanguinity (F_IT_), as well as the coefficient of genetic differentiation (F_ST_) and the endogamy coefficient (F_IS_), and a Factorial Analysis of Correspondence was also made using the program GENETIX v 4.05 [19]. The Genetic Distance was calculated (D_A_) [23] in the program POPULATIONS [24]. With the values of distance obtained, a phylogenic network of union of neighbors using the program SPLITSTREE [25] was made to represent the distance among the breeds.

### 2.5. Genetic Structure

For the analysis of the genetic substructure among individuals of the Oaxacan Creole sheep, a Bayesian algorithm was made with the program STRUCTURE v 2.1 [26] that uses a model based on the chain method of Monte Carlo Markov (MCMC) for estimating the distribution of coefficient of mixture of each individual (q). A total of 100,000 iterations of Burn-in were made with 300,000 iterations of a chain of MCMC.

## 3. Results

### 3.1. Intra-Species Genetic Diversity

The population analyzed of the Oaxacan Creole sheep presented a polymorphic character with respect to the allelic frequencies corresponding to the 41 microsatellites analyzed. A total of 205 alleles were found, with a minimum of two in the markers ETH010 and ILSTS008, and a maximum of nine in the marker OarCP49, showing a moderate allelic diversity in the population (Table 2).

In the analysis of genetic variability (Table 3), the Oaxacan Creole sheep presented a mean number of alleles of 5, an effective number of 3.327, expected values of heterozygosity (He) minimum 0.25 for ET010 and maximum of 0.963 for TGLA122, with a mean of 0.685; an observed heterozygosity (Ho) minimum of 0.0223 in ETH010 and maximum of 0.841 in OarCP49, with a mean of 0.671 by a direct recount.

The values of Polymorphic Information Content (PIC) in the population presented 8 markers with a value less than 0.5 and 33 with a value above 0.5 considered very informative for the characterization of genetic variability. The marker ETH010 is the only uninformative marker showing a PIC of < 0.25.

The value for the frequency of the coefficient of consanguinity (F_IS_) had a range of −0.301 (BM6526) to 0.332 (SPS115) with a mean of −0.022 (−0.087–0.002), utilizing 10,000 re-samplings and with a confidence interval of 95%, manifesting that this population does not have a significant deviation of Hardy–Weinberg equilibrium. When the Bonferroni correction was made, it was found that the markers BM6506, BM6526, CD5, ETH010, INRA063, TGLA053, and TGLA122 have a significant defect (*p* < 0.05) of homozygotes with negative values of F_IS_, presenting a deviation of the Hardy–Weinberg equilibrium.

### 3.2. Interracial Genetic Diversity

Parameters of genetic diversity of the 28 breeds are compared in Appendix A. The Oaxacan Creole sheep, when compared with native Spanish breeds, American Creoles, African breeds, and breeds more common in México, through a Factorial analysis of Correspondence, showed an elevated genetic differentiation, presenting statistical values of F_IS_ = 0.0774 (0.05353–0.10389), F_IT_ = 0.16993 (0.14598–0.19770), and F_ST_ = 0.10028 (0.09159–0.10917). In the Factorial Correspondence Analysis (Figure 2), the Oaxacan Creole sheep present a higher genetic distancing with the sheep of African origin (axis 2), followed by the hair sheep and being closer to the creoles of American and Spanish breeds (axis 1).

Analyzing the genetic distance (D_A_) and the coefficients of genetic differentiation with respect to other populations (F_ST_) (Appendix A), the Oaxacan Creole presents the lowest values with some creole sheep of America (D_A_ = 0.122–0.198; F_ST_ = 0.114–0.174), Africa (D_A_ = 0.194–0.163; F_ST_ = 0.140–0.175), and the highest values with the creole of Brazil (D_A_ = 0.247; F_ST_ = 0.218) and Palmera of Spain (D_A_ = 0.248; F_ST_ = 0.220).

The representation of D_A_ genetic distances through a neighbor-net graphic shows that the Oaxacan Creole sheep are grouped in the same branch as the Argentine Creole and share the same trunk of origin with the Creole of Mexico, Spanish Merino, and Chilean Merino (Figure 3), even though their genetic distances are high. Furthermore, interestingly, it is observed that it has a high genetic distance with the creole sheep of Chiapas, Mexico, which is the most popular creole sheep in Mexico.

### 3.3. Genetic Structure of the Oaxacan Creole Sheep

To determine the genetic substructure in the population of Oaxacan Creole sheep, efficiently assign individuals within the flock and that do not present any degree of crossbreeding with other populations, an analysis was made in the program STRUCTURE v 2.1, using a Bayesian algorithm of the program that calculates the distribution posteriori of each coefficient of a mixture of each individual (q) (Figure 4). This analysis shows each individual as a vertical bar and each color represents the proportion, in percentage, of the genome of the populations analyzed, making it possible to identify from which ancestral population the Oaxacan Creole sheep is from.

When two populations are analyzed (K2) (African vs. Merino), the Oaxacan Creole is grouped with those that descend from the Merino. The analysis of K3 shows the difference from the Spanish Merino and those that carry a greater genetic load of their ancestors (dark blue). When utilizing 10 populations (K10), more than 85% of the individuals of the Oaxacan Creole were assigned to the same group (data not shown). When compared with 16 populations, it is separated as an independent group, even from the other Mexican creoles. In the analysis of 25 (K25) (statistically optimum number) and 28 (K28) populations, 86.8% of the individuals are assigned to the same group, where no subdivision or substructure of the population is observed. Therefore, it can be considered a homogeneous flock without a populational substructure or genetic mixing with the rest of the breeds used.

## 4. Discussion

Sheep breeding in Mexico, in order to satisfy the productive, economic, and social needs of the producers and the population, has opted for improving flocks through the introduction of improved breeds and has carried out crossbreeding with no established control [27]. However, these practices have caused an erosion in the native genetic resources, increasing diversity and provoking their extinction [4,5,6,7]. Presently, programs have been established for the identification and conservation of native zoogenic resources for productive species [28,29]. One of the fundamental tools for this purpose is the molecular markers, principally the microsatellites, which facilitate identification, estimation of genetic diversity, detection of genes with qualitative features, and assisted selection [30].

In the present study, 29 sheep of the Oaxacan Creoles population, corresponding to one-third of the entire population, were genetically characterized, utilizing 41 microsatellites recommended by the FAO/ISAG for sheep [31]. These organizations recommend the minimum use of 25 markers for establishing genetic intra and interracial relationships of domestic animals. According to Nei et al. [23], the standard error of genetic diversity is lower when more *loci* are used. Furthermore, it is considered that the optimum sample size is 30, thus this study has a high degree of reliability due to the number of individuals and markers employed.

### 4.1. Intra-Racial Genetic Diversity

It was found that the population has a mean of alleles (N_AM_) of 5 and an effective number (Ae) of 3.327, parameters which in conjunction with heterozygosity, which indicate the genetic variability within a population. The value of N_AM_ is low compared with that of the sheep breeds of Colombia (14.27) [11], of China (13.59) [32], and of the creole of Chiapas (14.19) [9], but it is similar to that of the creole breeds of Argentina (Corrientes: 7.53; Santiago del Estero: 6.16; Salta: 6.7; Buenos Aires: 5.73) [33], to the creole of Paraguay (7.25) [34], to Italian breeds (Cornigliese: 6,64; Bergamasca: 5.56), and to the Spanish Merino (7.41) [35]. The values of Ae are similar to those found by Peña et al. [33] in 4 populations of Argentine creole sheep (Corrientes: 4.078; Santiago del Estero: 4.036; Salta: 3.856; Buenos Aires: 3.193) and to those reported in 13 breeds of sheep (3.64–4.43) of Colombia by Ocampo et al. [11].

The heterozygosity in the Oaxacan Creole is considered from medium to high according to what was reported by Martínez [33], given that it showed a mean of He = 0.686 and Ho = 0672, values similar to those presented by the creole of Chiapas, Mexico (He = 0.624–0712; Ho−0.606–0.666) [9], to the creoles of Argentina (He = 0.676; Ho = 0.685 [33] and the Spanish Merino (He = 0.661; Ho = 0.620 [36]. On the other hand, it is lower than the creole breed of Paraguay (He = 0.73) [34]. In addition, the Oaxacan Creole in the estimation of endogamy level [22] had a mean value of PIC = 0.609 and a mean consanguinity of F_IS_ = 0.022, showing polymorphic values of moderate to high and a moderate genetic diversity [14], lower than those present in Colombian sheep (PIC = 0.74; F_IS_ = 0.09) [11], but similar for the creole sheep of Chiapas, Mexico (F_IS_ = 0.026–0.076) [9], the Argentinian creoles (F_IS_ = 0.030 [33], and the Spanish Merino (F_IS_ = 0.074). Therefore, it can be concluded that the Oaxacan Creole is a population with a moderate intra-racial diversity without a significant deviation from the Hardy–Weinberg equilibrium, similar to other creole breeds of America and the Spanish Merino. Furthermore, the presence of markers with values of F_IS_ very far from zero (reduction of heterozygosity) in the Oaxacan Creole sheep may be due to the low number of individuals in the population, which favors endogamy, added to the lack of planning of crosses between unrelated individuals.

### 4.2. Inter-Racial Genetic Diversity

Genetic diversity among populations can be analyzed using two variables, the statistic of the coefficient of genetic differentiation (F_ST_) and the measurement of the genetic distance among them (D_A_) [37]. Estimating these values makes it possible to distinguish how genetically isolated a population is, which infer its phylogeny and generate strategies for its conservation.

The Oaxacan Creole sheep, in the analysis of genetic differentiation with the 27 populations utilized, showed a mean value of F_ST_−0.1003. This is similar to what was reported by Quiroz et al. [9] for the creole of Chiapas (FST = 0.1) and to that reported between the Argentinian creoles Santiago de Estero and Buenos Aires (0.104) [33]. According to the Wright classification [38], this value is contemplated as a moderate degree of genetic differentiation; but according to what was reported for domestic sheep breeds [9], it is considered that the Oaxacan Creole has a high degree of genetic differentiation in comparison to what was found by Rendón et al. [39] for Spanish breeds (FST = 0.07), to that of Merino breeds of different countries (0.03) reported by Diez-Tascón et al. [36], and to that estimated among the Colombian creoles (0.01–0.047) [13].

The analysis of the genetic relationships among the 28 populations in a neighbor-net built with D_A_ genetic distances showed the separation of 4 groups: grouping the Oaxacan Creole with another creole of Mexico, the Chilean Merino, the Spanish Merino, and with the lowest D_A_, with the Argentinian creole. This analysis makes it possible to suggest that the Oaxacan Creole conserves certain ancestry to the breeds introduced during the colonization [40], and by its D_A_ with the Spanish Merino and the Chilean Merino. On the other hand, it is grouped in the same phylogenic trunk as the Argentinian Creole and to the other creole of Mexico, suggesting the same origin, although with elevated genetic differences, and an origin different from that of the creole of Chiapas. This diversity or genetic differentiation of the Oaxacan Creole could be due to its geographic localization and to the management established by the producer, resulting in reproductive isolation, thus allowing the conservation of its genetic characteristics.

### 4.3. Genetic Structure

This analysis provides information for estimating the proportion that each individual or population carries the genome of its parents or ancestors, making it possible to distinguish if there is a mixing of breeds or not in order to assign individuals or homogeneous populations. This analysis helped to confirm what was observed in the dendrogram of the union of neighbors, revealing that the Oaxacan Creole sheep is a homogeneous population, with the ancestral influence of the Merino; without a populational substructure or mixing of its genome with any of the 27 populations analyzed, even with the other populations of the country, thus separating them as an independent breed.

Finally, the data obtained show that the sheep flock subjected to study in the region of the High Mixteca of Oaxaca is a creole breed different from those analyzed, maintaining its genetic merit. Therefore, its recognition is proposed as Chocholteca Creole breed for the zone in which it has been conserved and in honor of the ethnic group which inhabits this region. Furthermore, this recognition will help to promote its conservation and to implement populational recovery programs in conjunction with the producer and his cultural system.

## 5. Conclusions

The Chocholteca Creole sheep has been maintained as an isolated breed, conserving genetic characteristics since it was introduced to America, and does not present features of crossbreeding with the new lines of improved sheep or other creoles of America, or breeds of Spain and Africa.

This creole sheep has in its population a moderate–high genetic diversity, with a homogeneous structure, and does not present a subdivision or genetic substructure.

It can be considered as a creole breed different from the others of the American continent and Spain, establishing itself as an important new zoogenic resource. Therefore, it is necessary to establish reproductive management for its conservation.

## Figures and Tables

**Figure 1 animals-11-01172-f001:**
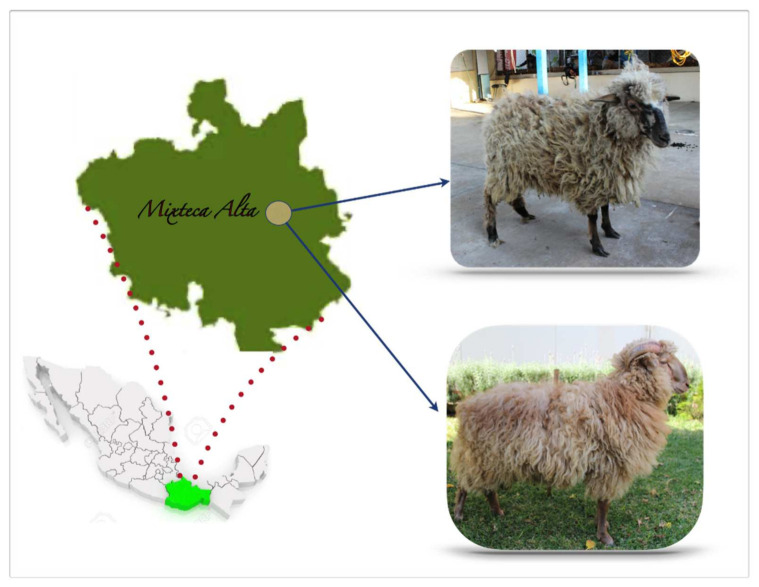
Oaxacan Creole sheep and their geographic location. A female is shown on the right (above) and a male (below) of the Oaxacan Creole sheep. The Chocholteca Creole Sheep has a great adaptive capacity to cold climate and pastures with low fodder. In an extensive grazing system, adult males and females reach 33.75 kg and 26.5 kg weight, respectively; if the feeding system improves, it is possible to find males of 60 kg and females of 38 kg. Males have 64 cm height at withers, 9 cm more than females. They can reproduce throughout the year, but it is possible to detect seasonality in the rainy season. They are medium-sized with a straight facial profile. They have long, open wool that invades the forehead. There are two types differentiated by the layer color: white and cardinal. White sheep with brown coffee spots on limbs and face are predominant in the herd. A small number of specimens show a cardinal layer (black and white hairs) with black spots on the face and limbs. Males have two horn types: athropy and spiral-shaped. Females lack horns.

**Figure 2 animals-11-01172-f002:**
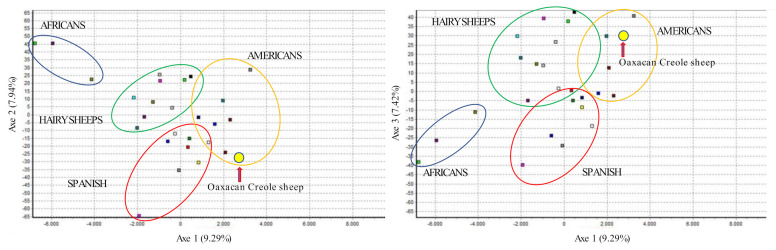
Factor of correspondence in the 28 sheep studied. The Oaxacan Creole sheep is shown in the yellow circle and indicated with a red arrow.

**Figure 3 animals-11-01172-f003:**
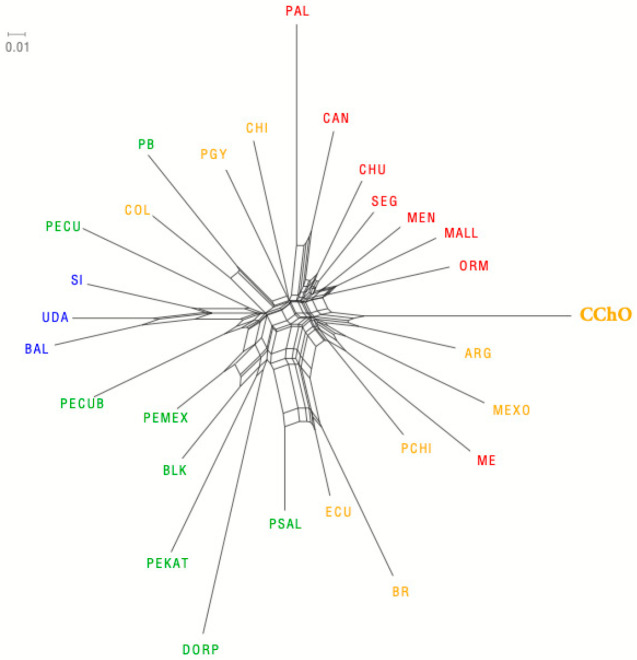
Representation of the DA genetic distances in a neighbor-net between the Oaxacan Creole and 27 other sheep populations. In yellow, Creole breeds; in red, Spanish breeds; in blue, African breeds; in green, Hairy sheep. ARG: Argentine Creole, BAL: Balami, BLK: Blackbelly, BR: Brazil Creole, CAN: Canarian, CChO: Creole Chocholteca Oaxaca, CHI: Chiapas Creole, CHU: Churra, COL: Colombian Creole, DORP: Dorper, ECU: Ecuador Creole, MALL: Mallorquina, ME: Merino Español, MEN: Menorquina, MEXO: México, ORM: Roja Mallorquina, PAL: Palmera, PB: Pelibuey, PCHI: Merino Chileno, PECU: Pelibuey Ecuador, PECUB: Pelibuey Cuban, PGY: Creole Paraguay, PEKAT: Katahdin, PEMEX: Pelibuey Mexicano, PSAL: Pelibuey El Salvador, UDA: UDA.

**Figure 4 animals-11-01172-f004:**
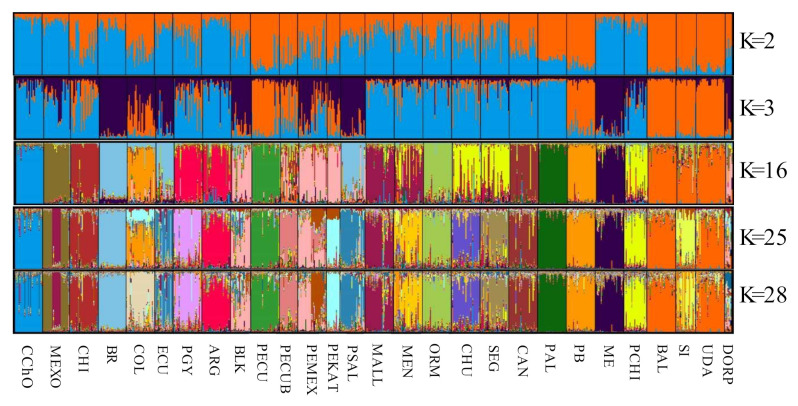
Genetic structure of the 28 sheep populations analyzed. The figure shows the graphic representation of the populational grouping when it is compared with a different number of populations (K): K = 2, K = 3, K = 16, K = 25 (optimum) and K = 28. In the X-axis the populations occupied are numbered (Table 1), where the 1 is the Oaxacan Creole.

**Table 1 animals-11-01172-t001:** Sheep breeds used in this study. The table shows the number of individuals used, their provenance, and the acronym for each one.

	Breed	Acronym	Provenance	*n*
1	Creole Oaxaqueño	CChO	México	29
2	México	MEXO	México	28
3	Borrego de Chiapas	CHI	México	30
4	Criollo de Brasil	BR	Brasil	29
5	Criollo Colombiano	COL	Colombia	30
6	Criollo Ecuador	ECU	Ecuador	19
7	Criollo Paraguay	PGY	Paraguay	30
8	Criollo Argentina	ARG	Argentina	30
9	Blackbelly	BLK	México	21
10	Pelibuey Ecuador	PECU	Ecuador	30
11	Pelibuey Cubano	PECUB	Cuba	19
12	Pelibuey Mexicano	PEMEX	México	30
13	Katahdin	PEKAT	México	14
14	Pelibuey El Salvador	PSAL	El Salvador	26
15	Mallorquina	MALL	Spain	30
16	Menorquina	MEN	Spain	30
17	Roja Mallorquina	ORM	Spain	30
18	Churra	CHU	Spain	30
19	Segureño	SEG	Spain	30
20	Canaria	CAN	Spain	30
21	Palmera	PAL	Spain	30
22	Pelibuey	PB	Spain	30
23	Merino Español	ME	Spain	30
24	Merino Chileno	PCHI	Chile	24
25	Balami	BAL	Nigeria	30
26	Sidaun	SI	Sahara	21
27	UDA	UDA	Nigeria	30
28	Dorper	DORP	South Africa	7

**Table 2 animals-11-01172-t002:** Allelic frequencies of the 41 microsatellites in the population of Oaxacan Creole sheep.

*Locus*	Allele	Frequency	*Locus*	Allele	Frequency	*Locus*	Allele	Frequency
BM1818	1	7.14	ILSTS008	1	57.14	McM527	1	16.67
	2	1.79		2	42.86		2	25
	3	26.79	ILSTS011	1	23.91		3	16.67
	4	3.57		2	13.04		4	16.67
	5	21.43		3	63.04		5	12.5
	6	16.07	ILSTS087	1	29.63		6	12.5
	7	19.64		2	22.22	OarAE129	1	8.93
	8	3.57		3	12.96		2	48.21
BM1824	1	26.09		4	11.11		3	32.14
	2	19.57		5	24.07		4	10.71
	3	4.35	INRA005	1	35.42	OarCP20	1	10.34
	4	50		2	29.17		2	32.76
BM6506	1	32.76		3	8.33		3	56.9
	2	34.48		4	14.58	OarCP34	1	54.17
	3	6.9		5	12.5		2	6.25
	4	25.86	INRA006	1	56.9		3	12.5
BM6526	1	56.9		2	27.59		4	27.08
	2	1.72		3	3.45	OarCP49	1	5.56
	3	17.24		4	12.07		2	12.96
	4	20.69	INRA023	1	58.93		3	29.63
	5	3.45		2	17.86		4	18.52
BM8125	1	20.69		3	16.07		5	11.11
	2	18.97		4	5.36		6	1.85
	3	8.62		5	1.79		7	5.56
	4	51.72	INRA035	1	6.9		8	12.96
CD5	1	28		2	60.34		9	1.85
	2	42		3	12.07	OarFCB11	1	8.62
	3	10		4	5.17		2	5.17
	4	8		5	15.52		3	41.38
	5	2	INRA049	1	53.45		4	8.62
	6	8		2	34.48		5	6.9
	7	2		3	10.34		6	29.31
CSRD247	1	3.45		4	1.72	OarFCB20	1	6.82
	2	27.59	INRA063	1	62.5		2	20.45
	3	39.66		2	3.57		3	31.82
	4	6.9		3	14.29		4	4.55
	5	1.72		4	5.36		5	36.36
	6	20.69		5	10.71	OarFCB304	1	70.69
CSSM66	1	1.72		6	3.57		2	3.45
	2	6.9	INRA132	1	8.33		3	25.86
	3	24.14		2	20.83	RM006	1	51.72
	4	24.14		3	10.42		2	1.72
	5	3.45		4	4.17		3	31.03
	6	24.14		5	16.67		4	15.52
	7	15.52		6	2.08	SPS113	1	12.96
D5S2	1	36.54		7	29.17		2	14.81
	2	7.69		8	8.33		3	25.93
	3	38.46	INRA172	1	7.41		4	46.3
	4	17.31		2	70.37	SPS115	1	2.08
ETH010	1	87.5		3	1.85		2	33.33
	2	12.5		4	20.37		3	4.17
ETH225	1	46.3	MAF065	1	11.54		4	58.33
	2	48.15		2	7.69		5	2.08
	3	3.7		3	34.62	TGLA053	1	22.92
	4	1.85		4	30.77		2	4.17
HSC	1	13.64		5	11.54		3	18.75
	2	4.55	MAF214	1	11.54		4	2.08
	3	15.91		2	46.15		5	2.08
	4	6.82		3	26.92		6	10.42
	5	22.73		4	15.38		7	39.58
	6	27.27	McM042	1	4.35	TGLA122	1	27.78
	7	9.09		2	39.13		2	12.96
ILSTS005	1	12.96		3	30.43		3	33.33
	2	31.48		4	4.35		4	20.37
	3	16.67		5	21.74		5	5.56
	4	11.11				TGLA126	1	6.9
	5	27.78					2	27.59
							3	20.69
							4	18.97
							5	6.9
							6	15.52
							7	3.54

**Table 3 animals-11-01172-t003:** Genetic results of the microsatellites in the Oaxacan Creole sheep. The table shows the values obtained for the number of alleles detected and expected, the expected and observed heterozygosity, the values of the coefficient of endogamy, and the deviation from the Hardy–Weinberg equilibrium.

Microsatellite	N° Allele	Ae	He	Ho	PIC	FIS	FIS IC	HWEd
ETH010	2	1.280	0.25	0.223	0.195	−0.125	(−0.244–−0.037)	1
OarFCB304	3	1.761	0.31	0.44	0.364	0.29805	(−0.050–0.612)	0.0315 *
SPS115	5	2.203	0.375	0.558	0.468	0.33226	(−0.031–0.633)	0.0256 *
ILSTS008	2	1.960	0.429	0.499	0.37	0.14286	(−0.242–0.496)	0.6978
INRA172	4	1.843	0.444	0.466	0.41	0.04733	(−0.181–0.225)	0.0082 *
RM006	4	2.576	0.517	0.623	0.542	0.1716	(−0.094–0.414)	0.1545
BM8125	4	2.827	0.517	0.658	0.596	0.21642	(−0.085–0.479)	0.0457
ETH225	4	2.233	0.519	0.563	0.451	0.07965	(−0.270–0.389)	0.8494
ILSTS011	3	2.120	0.522	0.54	0.467	0.03473	(−0.282–0.348)	0.7858
INRA049	4	2.406	0.552	0.595	0.508	0.07342	(−0.191–0.304)	0.007
INRA023	5	2.450	0.571	0.603	0.548	0.05263	(−0.134–0.204)	0.0466 *
INRA035	5	2.438	0.586	0.6	0.555	0.02359	(−0.224–0.252)	0.5145
OarCP20	3	2.264	0.621	0.568	0.48	−0.09446	(−0.379–0.174)	0.665
CSRD247	6	3.541	0.621	0.73	0.67	0.15223	(−0.081–0.357)	0.0043 *
OarAE129	4	2.815	0.643	0.656	0.583	0.02115	(−0.212–0.231)	0.4807
INRA006	4	2.406	0.655	0.595	0.523	−0.10373	(−0.310–0.062)	0.0156
D5S2	4	3.152	0.692	0.696	0.623	0.00552	(−0.235–0.220)	0.4133
BM1824	4	2.792	0.696	0.656	0.582	−0.06184	(−0.383–0.226)	0.1587
SPS113	4	3.122	0.704	0.693	0.628	−0.01646	(−0.266–0.204)	0.1321
ILSTS87	5	4.459	0.704	0.79	0.739	0.11151	(−0.120–0.316)	0.6321
OarCP34	4	2.589	0.708	0.627	0.556	−0.13333	(−0.431–0.148)	0.3552
INRA005	5	3.932	0.708	0.762	0.705	0.07126	(−0.195–0.304)	0.1144
INRA063	6	2.337	0.714	0.582	0.542	−0.23147	(−0.366–−0.114)	0.8619
HSC	7	5.408	0.727	0.834	0.79	0.13066	(−0.108–0.327)	0.0015 *
INRA132	8	5.460	0.75	0.834	0.793	0.10293	(−0.098–0.277)	0.0864
OarFCB11	6	3.579	0.759	0.733	0.679	−0.03529	(−0.249–0.149)	0.0817
MAF065	6	4.024	0.769	0.766	0.713	−0.00402	(−0.192–0.158)	0.4646
ILSTS005	5	4.288	0.778	0.781	0.729	0.00456	(−0.194–0.178)	0.0044 *
McM042	5	3.369	0.783	0.719	0.649	−0.09091	(−0.318–0.110)	0.7442
BM6526	5	2.514	0.793	0.613	0.552	−0.30101	(−0.448–−0.174)	0.4987
MAF214	4	3.101	0.808	0.691	0.625	−0.17318	(−0.383–0.024)	0.5427
OarFCB20	5	3.546	0.818	0.735	0.668	−0.11669	(−0.327–0.072)	0.6359
BM1818	8	5.262	0.857	0.825	0.783	−0.04013	(−0.194–0.082)	0.0793
BM6506	4	3.357	0.862	0.714	0.644	−0.21107	(−0.417–−0.028)	0.0482 *
CSSM66	7	4.875	0.862	0.809	0.764	−0.06707	(−0.228–0.075)	0.6016
CD5	7	3.592	0.88	0.736	0.682	−0.2	(−0.369–−0.048)	0.9183
OarCP49	9	5.718	0.889	0.841	0.804	−0.05852	(−0.192–0.045)	0.0426 *
TGLA126	7	5.273	0.897	0.825	0.784	−0.08901	(−0.233–0.035)	0.0247 *
McM527	6	5.647	0.917	0.84	0.798	−0.09287	(−0.234–0.037)	0.5744
TGLA053	7	3.879	0.958	0.758	0.704	−0.27163	(−0.419–−0.166)	0.1518
TGLA122	5	4.006	0.963	0.765	0.708	−0.26592	(−0.374–−0.180)	0.1761
Average	5	3.327	0.685	0.671	0.609	−0.022	(−0.087–0.002)	

Ae: number of alleles expected, He: unbiased expected heterozygosity, Ho: observed heterozygosity, PIC: polymorphic information content, FIS values of the coefficient of endogamy, FIS IC: confidence interval, HWEd: deviation from the Hardy–Weinberg equilibrium. * *p* < 0.05.

## Data Availability

The data in this study was microsatellite analysis by using known specific primers. No new data were created on this study. Data sharing is not applicable to this article.

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
