# Peer review of "Genetic Characterization of a Sheep Population in Oaxaca, Mexico: The Chocholteca Creole"

_animals, 2021, doi:10.3390/ani11041172_

Round 1

Reviewer 1 Report

This is a valuable contribution to the scientific literature since it describes a sheep population that has not been studied before. The sample size is not big, but is adequate for the analyses made. The methods used for the genetic diversity analyses are appropriate. The description of the methods, results and discussion is generally good but I have a few comments, listed below:

In general: breed is more commonly used than race in the context of livestock species.

Abstract: First sentance. The word ” miscegenation ” is not really an appropriate term for livestock. Please rephrase the sentance.

I section 2.1 it would be better to also have a short phenotypic description of the sheep. How big are they? Which colour(s) do they have? Doo they have horns? How many lambs do they typically get? No detailed description is required, but when reading a paper about a sheep population I would like to get some idea of which type of sheep it is. It is very good that there are photos I Figure 1, but do all sheep in this population look like that or are there some variation? The description of the shhep could be either in the introduction or in this section.

Table 1: South Africa is the correct English name of the country where the Dorper sheep comes from.

Author Response

the reply comments are in the pdf file

Reviewer 2 Report

This is an interesting study that makes a definite contribution to the knowledge base relating to the genetic characterization of the Oaxacan Creole sheep. First of all, the present manuscript is well designed, the methods are well described and the results are clear. The topics are also very interesting for the veterinary and animal science fields.

 However, there are some concerns which, in my opinion, may improve the intelligibility of this paper for the readers.

Minor comments

Authors should change the word “race” to “breed” in the whole text.

Results

Section 3.1.

In the text, the authors write that the average value of the Ho parameter is 0.765, however, in table 3 the mean value of this parameter is 0.671. Where does this difference come from?

Section 3.2.

I suggest transferring table 4 to the Supplements files and slightly modify (enlarge) it. This table contains a lot of numerical data and is hardly legible in its current form.

Author Response

the reply comments are in the pdf file

Reviewer 3 Report

In my view, this paper presents important information about Creole sheep population from Oaxaca, Mexico. The results of this study could be used in the breeding and conservation programs of Creole sheep in Mexico. I think this paper can be accepted after a major revision.

My suggestions:

  1. The title of the manuscript should be changed. I suggest: Genetic characterization of the Chocholteca Creole sheep population from Oaxaca, Mexico.
  2. The word race should be replaced with breed.
  3. It should be noted that currently whole genome SNP analysis is more efficient tool than microsatellites and why the authors think that STR analysis is appropriate in their study.
  4. Instead of Nei’s Da, Reynolds and Fst distances I would recommend to use Jost’s D genetic distances (Jost L. 2008. GST and its relatives do not measure differentiation. Mol Ecol 17: 4015-4026.) which were designed for microsatellites. It could be done in the R package diveRsity. The input file for this package is GENEPOP which the authors already have, so I don’t think this will be difficult.
  5. The quality of figures (Figure 2, Figure 3, Figure 4) must be improved. They all become blurry after they are zoomed in. For example: the Neighbor-Net could be saved as PDF in the SplitsTree software and then converted to high quality (600 dps) PNG in inkscape program.
  6. In the Figure 4, the names of the breeds should be given instead of the numbers. It would be easier to understand the results.
  7. Geographic coordinates of the Oaxaca region should be checked. I assume it is 96° and not 19°. If it is written – “between”, then the coordinates indicating another border must be specified. But perhaps it is better to indicate the coordinates of the central part of the region.
  8. The studied individuals should also be checked for their relatedness. One individual from the pair of full sibs should be removed. The presence of closely related animals (full sibs) could bias the study results. For example, it could be done using software ML-Relate (https://www.montana.edu/kalinowski/software/)
  9. In the Table 1 Criollo sheep should be written as Creole
  10. As far as I know, an automatic sequencer ABI 3130XL (Applied Biosystem) allows to conduct a capillary electrophoresis instead of the polyacrylamide gel as the authors indicated. Please, clarify this.
  11. I recommend to include a table with genetic diversity of all the studied breeds. The following parameters are needed: observed heterozygosity (Ho), expected heterozygosity (He), inbreeding coefficient (Fis) and allelic richness (Ar). So it will be easier to compare Creole from Oaxaca to the other breeds.
  12. The authors should indicate the current census size of the Chocholteca Creole sheep population from Oaxaca.
  13. The abbreviations of the breeds should be the same as throughout the entire manuscript. Now in the Table 1 Chocholteca Creole sheep is given as CChO, in the Table 4 – Ccho, in the Figure 3 – MX.
  14. I suggest to move Table 2 and Table 3 as well as Figure 2 to Supplementary material.

Author Response

the reply comments are in the pdf file

Round 2

Reviewer 3 Report

The authors have greatly improved the manuscript, but I still have a number of remarks

  1. In the “2.3. Intra-racial genetic diversity” section

Instead of “by means of the chain method of Monte Carlo Marcov”, the authors would better write “by Markov chain Monte Carlo (MCMC) method”

  1. Figure 3. The names of all the breeds but Creole are missing. All the breeds’ names should be presented in the figure.

Instead of “dendrogram of union of neighbors” it would be correct “Neighbor-Net dendrogram”

  1. In Discussion, in the sentence “In the present study, 29 sheep of the Oaxacan Creoles population were genetically characterized” the authors should emphasize that one third of the sheep population was genotyped.
  2. The authors mention twice Reynolds distances in the Results. However, these distances were not calculated.
